# Genomic language model predicts protein co-regulation and function

## Abstract

Deciphering the relationship between a gene and its genomic context is fundamental to understanding and engineering biological systems. Machine learning has shown promise in learning latent relationships underlying the sequence-structure-function paradigm from massive protein sequence datasets; However, to date, limited attempts have been made in extending this continuum to include higher order genomic context information. Here, we trained a genomic language model (gLM) on millions of metagenomic scaffolds to learn the latent functional and regulatory relationships between genes. gLM learns contextualized protein embeddings that capture the genomic context as well as the protein sequence itself, and appears to encode biologically meaningful and functionally relevant information (e.g. enzymatic function). Our analysis of the attention patterns demonstrates that gLM is learning co-regulated functional modules (i.e. operons). Our findings illustrate that gLM's unsupervised deep learning of the metagenomic corpus is an effective and promising approach to encode functional semantics and regulatory syntax of genes in their genomic contexts and uncover complex relationships between genes in a genomic region.

## 1 Introduction

### 1.1 Background

Evolutionary processes result in the linkage between protein sequences, structure and function. The resulting sequence-structure-function paradigm has long provided the basis for interpreting vast amounts of genomic data. Recent advances in neural network (NN)-based protein structure prediction methods Jumper (2021); Baek (2021), and more recently protein language models (pLMs) Rives (2021); Elnaggar (2020); Madani (2023) suggest that data-centric approaches in unsupervised learning can represent these complex relationships shaped by evolution. To date, These models largely consider each protein as an independent and standalone entity. However, proteins are encoded in genomes, and the specific genomic context that a protein occurs in is also determined by evolutionary processes, where each gene gain, loss, duplication and transposition event is subject to selection and drift Wright (1948); Lynch & Conery (2003); Cordero & Polz (2014). These processes are particularly pronounced in prokaryotic genomes where frequent horizontal gene transfers (HGT) shape genomic organization and diversity Treangen & Rocha (2011); Shapiro (2012). Thus, there exists an inherent evolutionary linkage between genomic context and gene function Kountz & Balskus (2021), which can be explored by characterizing patterns that emerge from large metagenomic datasets.

Submitted to NeurIPS 2021 AI for Science Workshop.

## 1.2 Related works

Recent efforts to model genomic information have shown predictive power of genomic context in gene function Miller et al. (2022) and metabolic trait evolution Konno & Iwasaki (2023) in bacterial and archaeal genomes. However, both methods represent genes as categorical entities, despite these genes existing in continuous space where multidimensional properties such as phylogeny, structure, and function are abstracted in their sequences. On the other end of the spectrum of representations, there have been efforts to use unsupervised learning on nucleotide sequences to predict gene expression level Avsec et al. (2021) and detect regulatory motifs Avsec et al. (2021); Ji et al. (2021); Dalla-Torre et al. (2023); Nguyen et al. (2023). These models are largely trained and benchmarked on the human genome and focus on predicting gene regulation rather than function. Previous efforts to leverage diverse microbial sequences to model genome-scale information include GenSLMs Zvyagin et al. (2022), which is pretrained on codon-level representations of diverse bacterial and viral gene sequences and later fine-tuned on SARS-CoV-2 genomes. In order to learn generalizable gene-to-gene-context interactions across biology, a model needs to be pretrained on 1) diverse lineages of organisms, 2) rich and continuous representation of genes and 3) longer segments of genomes with multiple genes. To our knowledge, there has been no method that combines all three aspects of pretraining to learn genomic information across diverse lineages of biology (see summary of previous efforts in Table 1).

## 1.3 Genomic language modeling

In order to close the gap between genomic-context and gene sequence-structure-function, we developed the first, to our knowledge, genomic language model (gLM) that represents proteins using pLM embeddings that have been shown to encode relational properties Rives (2021) and structure information Lin (2023). Our model, based on the transformer architecture Vaswani et al. (2017), is trained using millions of unlabelled metagenomic sequences. We trained gLM with the masked language modeling Devlin et al. (2018) objective, with the hypothesis that its ability to attend to different parts of a multi-gene sequence will result in the learning of gene functional semantics and regulatory syntax (e.g. operons). Here, we report evidence of the learned contextualized protein embeddings and attention patterns capturing biologically relevant information.

# 2 Methods

## 2.1 Masked language modeling of genomic sequences

The genomic corpus was generated using the MGnifyRichardson (2023) dataset (released 2022-05-06 and downloaded 2022-06-07). First, genomic contigs with greater than 30 genes were divided into 30 gene non-overlapping subcontigs resulting in a total of 7,324,684 subcontigs with lengths between 15 and 30 genes (subcontigs < 15 genes in length were removed from the dataset). To model genomic sequences, we trained a 19-layer (954M parameter) transformer model (Fig. 1A) on seven million metagenomic contig fragments consisting of 15 to 30 genes from the MGnify Richardson (2023) database. Each gene in a genomic sequence is represented by a 1280 feature vector (context-free protein embeddings) generated by using ESM2 pLM Rives (2021), concatenated with an orientation feature (forward or backward). For each sequence, 15% of genes are randomly masked, and the model learns to predict the masked label using the context. Based on the insight that more than one gene can legitimately be found in a particular genomic context, we allow the model to make four different predictions and also predict their associated probabilities. Thus, instead of predicting their mean value, the model can approximate the underlying distribution of multiple genes that can occupy a genomic niche We assess the model's performance using a pseudo-accuracy metric, where a prediction is considered correct if it is closest to the masked protein in euclidean distance compared to the other proteins encoded in the sequence. Dataset used for training is available for download from the MGnify server: `http://ftp.ebi.ac.uk/pub/databases/metagenomics/`

Table 1: Comparison of gLM to previous efforts in modeling various aspects of biological sequences.

| | Multi-gene interaction | Continuous representation of genes | Generalizable across organisms | Self-supervised language model |
|---|---|---|---|---|
| gLM (this study) | ✓ | ✓ | ✓ (Metagenomic sequences with bias towards bacteria, archaea and viruses) | ✓ |
| pLMs Lin (2023); Elnaggar (2020); Madani (2023) (e.g. ESM2, ProtBert, ProGen) | ✗ | ✓ | ✓ | ✓ |
| Miller et al. (2022) | ✓ | ✗ | ✓ | ✗ |
| Enformer Avsec et al. (2021) | ✓ | ✓ | ✗ (Pretrained on human and mouse genomes only) | ✗ |
| DNABERT Ji et al. (2021) | ✗ (Max context length of DNABERT-6 is 3072 bp, which is not sufficient to include a median length (26,288 bp) human protein coding gene) | ✓ | ✗ (Pretrained on human genome) | ✓ |
| Nucleotide Transformer Dalla-Torre et al. (2023) | ✗ (Max context length is 6000 bp, which is not sufficient to include a median length (26,288 bp) human protein coding gene) | ✓ | ✗ (Heavily biased towards human genome) | ✓ |
| HyenaDNA Nguyen et al. (2023) | ✓ | ✓ | ✗ (Pretrained on human genome) | ✓ |
| GenSLM Zvyagin et al. (2022) Foundation model | ✗ (Single genes used for pretraining | ✓ | ✓ | ✓ |
| GenSLM-SARS-CoV2 genome model Zvyagin et al. (2022) | ✓ | ✓ | ✗ (fine-tuned on SARS-CoV2 genomes only) | ✓ |

peptide_database/2022_05/. Training and inference code and analysis scripts are available at https://github.com/y-hwang/gLM.

## 2.2 Enzyme Commission number prediction

Custom MGYP-Enzyme Commission (MGYP-EC) dataset was created by first searching (mmseqs261 with default setting) MGYPs against the "split30.csv" dataset previously used to train CLEAN Yu (2023). "split30.csv" dataset consists of EC numbers assigned to UniProt sequences clustered at 30% identity. Only MGYP hits with >70% sequences to "split30.csv" were considered and MGYPs with multiple hits with >70% similarity were removed. Test split was selected by randomly selecting 10% of "split30.csv" UniProt IDs in each EC category that map to MGYPs. EC categories with less than four distinct UniProt IDs with MGYP mapping were removed from the dataset, resulting in 253 EC categories. pLM (context-free) embeddings were calculated for each of MGYP with EC number assignment by mean-pooling the last hidden layer of its ESM2 embedding. gLM (contextualized) embeddings were calculated also for each layer by running inference without masking and subsequently extracting per-layer hidden representations for MGYPs with EC number assignments. Linear probing was conducted for these embeddings with a single linear layer. Linear

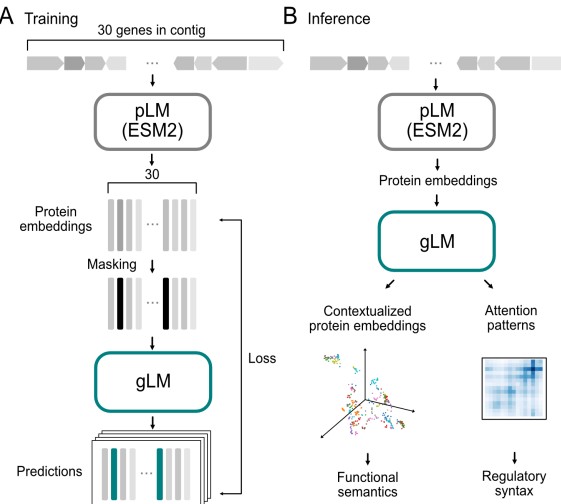

Figure 1: gLM training and inference schematics. A) For training, contigs (contiguous genomic sequences) containing up to 30 genes are first translated into proteins, which are subsequently embedded using a pLM encoder (ESM2). Masked inputs are generated by random masking at 15% probability and gLM (a transformer encoder) is trained to make four predictions for each masked protein, with associated likelihoods. Training loss is calculated on both the prediction and likelihoods. B) At inference time, inputs are generated from a contig using ESM2 output. Contextualized protein embeddings (last hidden layer of gLM) and attention patterns are used for various downstream tasks.

probes were trained with early stopping and batch size = 5000, and training results were replicated five times with random seeds to calculate error ranges.

### 2.3 Attention and operon analysis

Attention heads (n = 190) were extracted by running inference on unmasked subcontigs, and the raw attention weights were subsequently symmetrized. E.coli K12 RegulonDB Tierrafría (2022) was used to probe heads with attention patterns that correspond the most with operons. Pearson's correlation between symmetrized raw attentions and operons were calculated for each head. We trained a logistic regression classifier that predicts whether two neighboring genes belong to the same operon based on the attention weights across all attention heads corresponding to the gene pair.

## 3 Results

### 3.1 Model performance

We validate our model's performance on the Escherichia coli K-12 genome by excluding from training 5.1% of MGnify subcontigs in which more than half of the proteins are similar (>70% sequence identity) to E. coli K-12 proteins. The goal here is not to remove all E. coli K-12 homologs from the training, which would have removed a vast majority of training data as many essential genes are shared across organisms. Instead, our goal was to remove as many E.coli K-12-like genomic contexts (subcontigs) from training, which is more appropriate for the training objective. gLM achieves 71.9% in validation pseudo-accuracy and 59.2% in validation absolute accuracy. Notably, 53.0% of the predictions made during validation are with high confidence (with prediction likelihood > 0.75), and 75.8% of the high confidence predictions are correct, indicating gLM's ability to learn a confidence metric that corresponds to increased accuracy. We baseline our performance with a bidirectional LSTM model trained using the same language modeling task on the same training dataset, where validation performance plateaus at 28% pseudo-accuracy and 15% absolute accuracy. We ablate the use of pLM representations as input to gLM by replacing them with one-hot amino

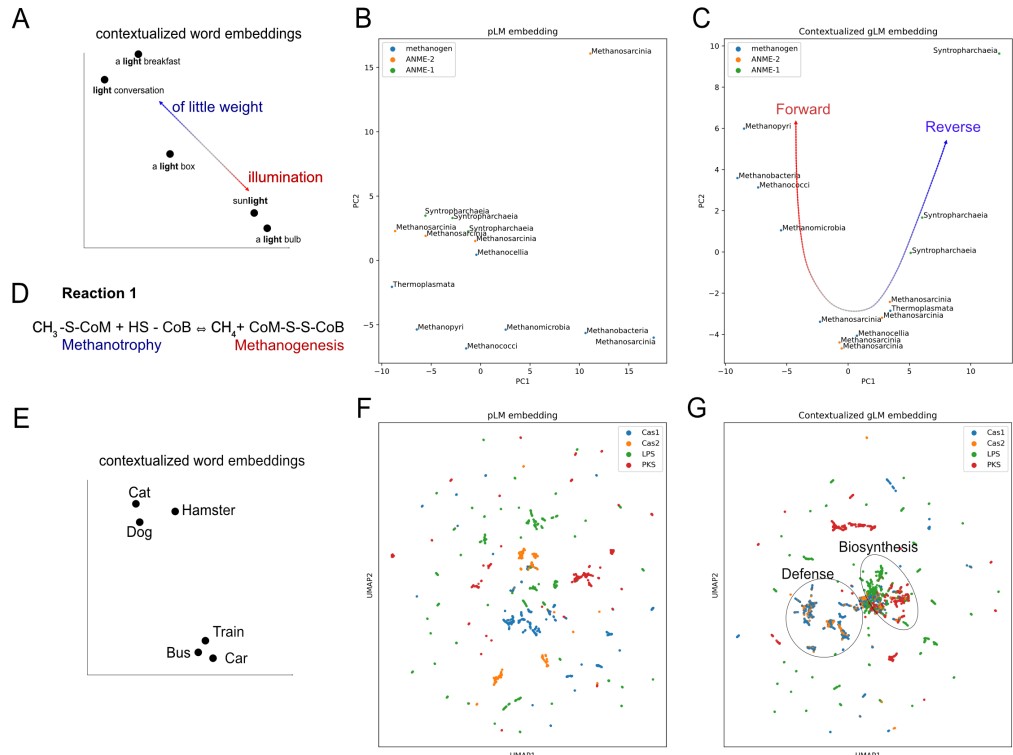

Figure 2: Contextualized protein embedding analysis and comparison with concepts in natural language modeling. **A**) A word's meaning upon contextualization varies across a continuous spectrum and can be ambiguous even with contextualization (e.g. double entendre). **B**) Input protein embeddings of McrA sequences in genomes, colored by metabolic classification of the organism (ANME, methanogen) based on previous studies and labeled by class-level taxonomy. **C**) Clustering of McrA sequences upon contextualization, with the likelihoods in the direction of Reaction 1 that the MCR complex carries out. **D**) Reaction 1, carried out by the MCR complex, either backward (Methanotrophy) or forward (Methanogenesis). **E**) Geometric relationship between contextualized protein embeddings based on the semantic closeness of words. **F**) Input (context-free) protein embeddings of Cas1, Cas2, lipopolysaccharide synthases (LPS) and polyketide synthases (PKS) showing clustering based on structural and sequence similarity. **G**) Clustering of contextualized protein embeddings where phage defense proteins cluster (Cas1 and Cas2) and biosynthetic gene products cluster (LPS and PKS).

acid representations and report performance equivalent to random predictions (3% pseudo-accuracy and 0.02% absolute accuracy).

## 3.2 Contextualized gene embeddings capture gene semantics

The mapping from gene to gene-function in organisms is not one-to-one. Similar to words in natural language, a gene can confer many different functions Jeffery (2018) depending on its context Miskei (2017), and many genes can confer similar functions (i.e. convergent evolution Gherardini et al. (2007), remote homology Ben-Hur & Brutlag (2003)).

We explored an ecologically important example of genomic "polysemy" (multiple meanings conferred by the same word) of methyl-coenzyme M reductase (MCR) complex (Fig. 2ABC). The MCR complex is able to carry out a reversible reaction (Reaction 1 in Fig. 2D), whereby the forward reaction results in the production of methane (methanogenesis) while the reverse results in methane oxidation (methanotrophy). We first examine the McrA (methyl-coenzyme M reductase subunit alpha) protein in diverse lineages of ANME (ANaerobic MEthane oxidizing) and methanogenic archaeal genomes. These archaea are polyphyletic and occupy specific ecological niches. Notably,

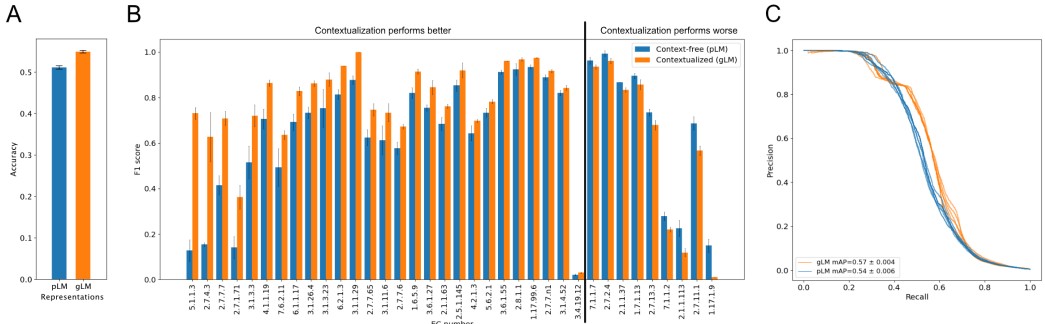

Figure 3: Contextualization of enzyme function. A) Linear probe EC classification accuracy for pLM (ESM2) representations and gLM (1st hidden layer) representations. B) F1-score comparisons of statistically significant (Benjamini/Hochberg corrected p-value < 0.05) differences in performance of pLM- and gLM-based EC number linear probes. EC classes are ordered with the largest gain with contextualization on the left to the largest loss with contextualization on the right. C) Precision-Recall curves of pLM- and gLM-based EC number linear probes.

similar to how a semantic meaning of a word exists on a spectrum and a word can have multiple semantically appropriate meanings in a context (Fig. 2B), the MCR complex can confer different functions depending on the context. Previous reports demonstrate capacities of ANME (ANME-2 in particular) carrying out methanogenesis Bertram (2013) and methanogens conducting methane oxidation in specific growth conditions Moran et al. (2007). The context-free ESM2 embedding of these proteins (Fig. 2E) shows little organization, with little separation between ANME-1 and ANME-2 McrA proteins. However, contextualized gLM embeddings Fig. 2C) of the McrA proteins show distinct organization where ANME-1 McrA proteins form a tight cluster, while ANME-2 McrA proteins form a cluster closer to methanogens (silhouette score after contextualization: 0.24; before contextualization:0.027). This organization reflects the phylogenetic relationships between the organisms that McrAs are found in, and reflect distinct operonic and structural divergence of MCR complexes in ANME-1 compared to those found in ANME-2 and methanogens Shao (2022). As proposed by Shao et al., the preferred directionality in Reaction 1 (Fig. 2G) in ANME-2 and some methanogens may be more dependent on thermodynamics.

We also demonstrate that contextualized gLM embeddings are more suitable for determining the functional relationship between gene classes. Analogous to how the words "dog" and "cat" are closer in meaning relative to "dog" and "train" (Fig. 2E), we see a pattern where Cas1 and Cas2 that appear diffuse in multiple subclusters in context-free protein embedding space (Fig. 2F) cluster in contextualized embedding space (Fig. 2G). This reflects their similarity in function (e.g. phage defense). This is also demonstrated in biosynthetic genes, lipopolysaccharide synthase (LPS) and polyketide synthase (PKS) genes clustering closer together in contextualized embedding space distinct from the Cas proteins (Fig. 2G). We quantitate this pattern with a higher silhouette score measuring phage defense and biosynthetic gene separation (gLM representation: 0.105±0.012, pLM representation: 0.078±0.011; paired t-test, t-statistic: 4.6, p-value = 0.001, n=10). Contextualized protein embeddings are therefore able to capture relational properties semantic information Reif (2019), where proteins that are more similar in their function appear in more similar genomic contexts.

### 3.3 Contextualization improves enzyme function prediction

To test the hypothesis that the genomic context of proteins can be used to aid function prediction, we evaluated how contextualization can improve the expressiveness of protein representations for enzyme function prediction. First, we generated a custom MGYP-EC dataset where the train and test data were split at 30% sequence identity for each EC class Yu (2023). Second, we apply a linear probe (LP) to compare the expressiveness of representations at each gLM layer, with and without masking the queried protein (Extended Data 8). By masking the queried protein, we can assess gLM's

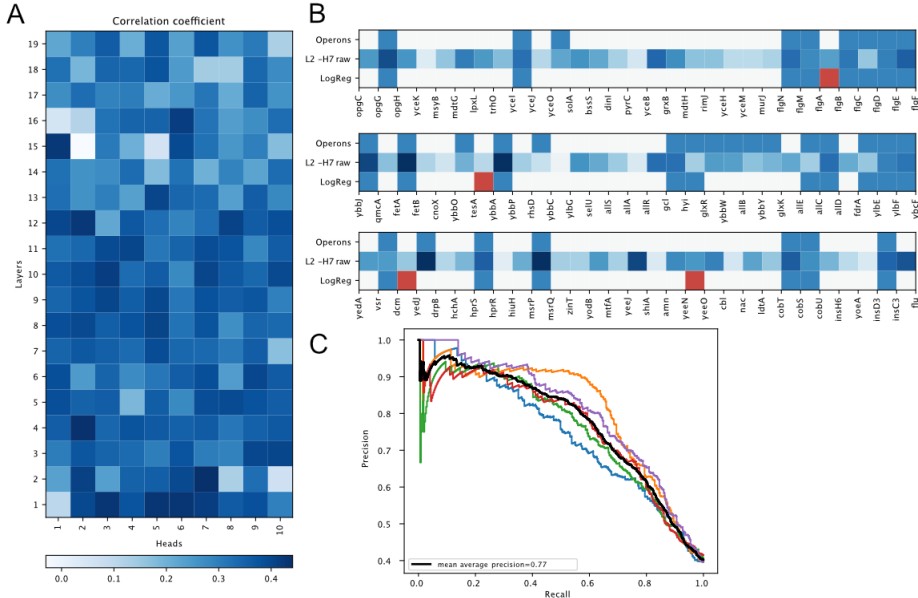

Figure 4: Attention analysis. A) Correlation coefficients (Pearson's rho) between attention heads across layers and operons. Darker color corresponds to stronger correlation with previously identified operons. Attention patterns of the second layer-seventh head [L2-H7] is most strongly correlated with the operons. B) Three random examples of contigs and predicted operonic relationship between neighboring proteins. Proteins are listed in the order they are encoded in the contig. Ground truth E. coli K-12 operons (top row), raw attention scores in the attention head [L2-H7] most correlated with operons (middle row) and logistic regression prediction using all attention heads (last row) where false positive predictions are marked in red. C) Five-fold cross-validation precision-recall curves of logistic regression trained using all operons and attention heads.

ability to learn functional information of a given protein, only from its genomic context, without the propagation of information from the protein's pLM embeddings. We observed that a large fraction of contextual information pertaining to enzymatic function is learned in the first six layers of gLM. We also demonstrate that context information alone can be predictive of protein function, reaching up to 24.4 ± 0.8% accuracy. In contrast, without masking, gLM can incorporate information present in the context with the original pLM information for each queried protein. We observed an increase in expressivity of gLM embeddings also in the shallower layers, with accuracy reaching up to 51.6 ± 0.5% in the first hidden layer. This marks a 4.6 ± 0.5% increase from context-free pLM prediction accuracy (Fig. 3A) and mean average precision (Fig. 3C) Thus, we demonstrate that information that gLM learns from the context is orthogonal to information captured in pLM embedding. We also observed diminishing expressivity in enzyme function information with deeper layers of gLM; this reflects the masked pretraining objective that is independent of enzyme function prediction task and is consistent with previous examinations of LLMs, where specific layers perform better than others for downstream tasks. Finally, to further examine the expressiveness of these representations, we compared per-class F1 score gains (Fig. 3B). We observe statistically significant differences in F1 scores (t-test, Benjamini/Hochberg corrected p-value < 0.05) between the two models in 36 out of 67 EC classes with more than ten samples in the test set. Majority (27 out of 36) of the statistical differences resulted in improved F1 score in LP trained on gLM representations.

### 3.4 Transformer's attention captures operons

The transformer attention mechanism models pairwise interaction between different tokens in the input sequence. Previous examinations of the attention patterns of transformer models in natural language processing (NLP) Rogers et al. (2020) have suggested that different heads appear to

specialize in syntactic functions. Subsequently, different attention heads in pLMs Vig (2020) have been shown to correlate to specific structural elements and functional sites in a protein. For our gLM, we hypothesized that specific attention heads focus on learning operons, a "syntactic" feature pronounced in in microbial genomes where multiple genes form regulatory modules. We used the E.coli K-12 operon database Salgado (2018) consisting of 817 operons for validation. gLM contains 190 attention heads across 19 layers. We found that heads in shallower layers correlated more with operons (Fig. 4A), with raw attention scores in the 7th head of the 2th layer [L2-H7] linearly correlating with operons with 0.44 correlation coefficient (Pearson's rho, Bonferroni adjusted p-value $< 1E-5$) (Fig. 4B). We further trained a logistic regression classifier using all attention patterns across all heads. This classifier predicted the presence of an operonic relationship between a pair of proteins in a sequence with mean average precision of 0.77 (Fig. 4C).

# 4 Discussion

The work presented here demonstrates and validates the concept of genomic language modeling. Taken together, gLM presents a highly promising direction for interpreting biology and we propose key areas for further development: First, the transformer architecture has shown to be successful in efficient scaling; in both natural language Kiros et al. (2014) and protein language processing Lin (2023), increasing the number of parameters in the model along with the training dataset size have been shown to lead to vastly improved performance and generalizability. Our model consists of 1B parameters which is at least a magnitude smaller compared to state-of-the-art pLMs. With further hyperparameter tuning and scaling, we expect better performance of the model. Second, our model currently uses protein-level pLM embeddings to represent proteins in the input. These embeddings are generated by mean-pooling the amino acid residue-level hidden states across the protein sequence, and therefore the residue specific information and synonymous mutation effects are likely obscured. Future iterations of the model could use raw residue-level or codon-level embeddings as input to allow modeling of residue-to-residue co-evolutionary interactions between proteins and synonymous mutation effects on gene function. Third, the task of reconstructing masked protein embeddings requires modeling a distribution over possible embeddings; our method approximates this distribution using a fixed number of predictions. Future work could improve upon this by using a generative approach, such as a diffusion or GAN model. This may allow for better prediction accuracy and greater generalizability for unseen datasets. Fourth, adding non-protein modalities (e.g. non-coding regulatory elements) as input to gLM may also greatly improve gLM's representation of biological sequence data, and can learn protein function and regulation conditioned upon other modalities Kiros et al. (2014). Finally, our model was trained largely on bacterial, archaeal and viral genomes, therefore, how this method can be adapted for eukaryotic genomes, especially those with extensive intergenic regions, remains to be further explored.

One of the most powerful aspects of the transformer-based language models is their potential for transfer learning and fine-tuning. We tested some of the capabilities of gLM and successfully showed that higher order biological information including gene function and regulation can be learned using genomic sequences. Our results highlight the importance of contextualization of biological data, particularly as we scale our modeling efforts from biomolecules to whole organisms. We propose the following promising future directions for applying gLM for advancing biological research. 1) Feature-based transfer learning for predicting protein function (e.g. Gene Ontology [GO] term, EC number), particularly those with limited sequence and structural homology. 2) Fine-tuning gLM for the protein-protein-interactome prediction task. 3) Using gLM features to encode genomic contexts as additional input for improved and contextualized protein structure predictions. In conclusion, genomic language modeling is a powerful tool to unbiasedly condense important biological information from full metagenomic sequences. Coupled with the advances in long-read sequencing, we expect a drastic increase in the input data quality, quantity and diversity. Genomic language modeling presents an avenue to bridge the gap between atomic structure and organismal function, and thereby brings us closer to modeling biological systems, discovering novel biology, and ultimately, manipulating biology with precision (e.g. genome editing, synthetic biology).

## 5   Acknowledgements

We would like to thank the EBI MGnify team for generating and maintaining the metagenome database. We would also like to thank Meta AI's ESM developers who made both the folded MGnify proteins structures and source-code openly available.

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
