# OpenReview forum: "Genomic language model predicts protein co-regulation and function"
_NeurIPS.cc/2023/Workshop/AI4Science — NeurIPS2023-AI4Science Poster_

### Meta-Review · Area_Chair_zZxP · 2023-10-27

**Recommendation:** Accept (Poster)
**Confidence:** 5

**Metareview:**

This paper proposes gLM, a genomic foundation model. gLM is able to capture the functional and regulatory information in the biological domain. Here are some comments that can be improved for the authors:
- The subfigures in Figure 2 can be further improved.
- The qualitative results in Figures 2-4 are interesting. However, a more in-depth analysis, especially the co-regulation part , would be interesting.